# Uncovering the Root of Hate Speech:
# A Dataset for Identifying Hate Instigating Speech

**Hyoungjun Park**     **Ho Sung Shim**     **Kyuhan Lee**

Department of Information Systems, Korea University Business School,
Seoul, Republic of Korea
{parkhj1111, hsshim9702, kyuhanlee}@korea.ac.kr

## Abstract

While many prior studies have applied computational approaches, such as machine learning, to detect and moderate hate speech, only scant attention has been paid to the task of identifying the underlying cause of hate speech. In this study, we introduce the concept of hate instigating speech, which refers to a specific type of textual posts on online platforms that stimulate or provoke others to engage in hate speech. The identification of hate instigating speech carries substantial practical implications for effective hate speech moderation. Rather than targeting individual instances of hate speech, by focusing on their roots, i.e., hate instigating speech, it becomes possible to significantly reduce the volume of content that requires review for moderation. Additionally, targeting hate instigating speech enables early prevention of the spread and propagation of hate speech, further enhancing the effectiveness of moderation efforts. However, several challenges hinder researchers from addressing the identification of hate instigating speech. First, there is a lack of comprehensive datasets specifically annotated for hate instigation, making it difficult to train and evaluate computational models effectively. Second, the subtle and nuanced nature of hate instigating speech (e.g., seemingly non-offensive texts serve as catalysts for triggering hate speech) makes it difficult to apply off-the-shelf machine learning models to the problem. To address these challenges, in this study, we have developed and released a multilingual dataset specifically designed for the task of identifying hate instigating speech. Specifically, it encompasses both English and Korean, allowing for a comprehensive examination of hate instigating speech across different linguistic contexts. We have applied existing machine learning models to our dataset and the results demonstrate that the extant models alone are insufficient for effectively detecting hate instigating speech. This finding highlights the need for further attention from the academic community to address this specific challenge. We expect our study and dataset to inspire researchers to explore innovative methods that can enhance the accuracy of hate instigating speech detection, ultimately contributing to more effective moderation and prevention of hate speech propagation online.

## 1   Introduction

Hate speech is a pressing issue in our society that requires significant attention. To this end, machine learning (ML) has gained a large amount of attention from both academia and industry as a valuable means to address hate speech (Davidson et al., 2017). In general, ML methods use features extracted from textual content, such as document embeddings and POS tags, with other behavioral information, such as social interactions, to enhance hate speech classification performance. Previous research (Burnap and Williams, 2015; Waseem and Hovy, 2016) has demonstrated the promising results of these approaches in identifying instances of hate speech.

Although the identification and moderation of hate speech hold significance per se, the paramount objective should lie in eliminating the underlying cause that gives rise to hate speech. For instance, within the context of online interactions, various forms of discourse exist that may not explicitly embody hatred itself but instead contribute to the provocation of hostility, aggression, and toxicity. These include behaviors like trolling. In this study, we refer to these behaviors as *hate instigating* behaviors. While it is crucial to identify and control hate instigating behaviors (or more specifically

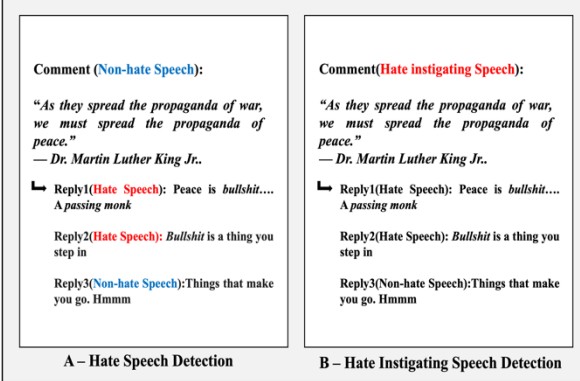

Figure 1: Comparison of hate speech and hate instigating speech detection in social media.

hate instigating speech) as an early preventive measure against the proliferation of hate speech, this issue has often received limited attention, particularly from a computational perspective.

Figure 1-A illustrates the traditional hate speech detection process, which evaluates texts, regardless of their conversational levels (i.e., comments and replies) within online platforms to determine the presence of hate speech. This approach focuses solely on identifying instances of hate speech without explicitly considering whether those instances incite or provoke hatred in others. On the other hand, in Figure 1-B, we introduce the concept of hate instigating comments. That is, in addition to the identification of hate speech, it highlights the examination of the likelihood of each comment provoking hatred of others. In Figure 1-B, although the comment itself does not exhibit any direct expressions of hatred (as in Figure 1-A), it serves as a comment that elicits hate-filled replies from others, which makes it a hate instigating comment. The identification of hate instigating comments holds significant practical implications as it serves as a crucial deterrent in mitigating the spread of hate speech. As an example, the content moderation workforce can strategically adjust their focus by allocating more resources to monitoring and addressing comment threads[1] that involve hate instigating aspects. By doing so, they can enhance hate moderation efficiency by reducing attention given to non-hate-instigating comments that have a lesser impact on the spread of hate speech.

In this study, we aimed to address the issue of hate instigating comments in online platforms by creating multilingual dataset[2], in both English and Korea, which can facilitate the development of computational algorithms for mitigating the spread of hate speech. Specifically, we have collected comments and replies on the videos from multiple YouTube channels operated by major US and Korean news media companies, i.e., CNN, Fox News, JTBC, and TVChosun. Then, human annotators manually labeled the data as hate speech, some of which are further classified as hate instigating speech. By incorporating hate instigator detection [3] alongside traditional hate speech detection, we aim to gain a more nuanced understanding of the dynamics and triggers that contribute to the propagation of hate speech within online communities. We expect this to promote the development of targeted interventions and proactive measures to disrupt the spread of harmful content. In the upcoming sections, we will examine the existing literature on hate speech detection and provide a comprehensive overview of our dataset generation process. Subsequently, we will apply various ML techniques to our dataset and discuss the practical implications of our findings.

## 2 Related literature

### 2.1 Hate speech detection datasets

In prior literature, many hate speech datasets have been suggested. These existing datasets have played a pivotal role in advancing research on hate speech detection by providing valuable resources for training and testing ML models. In a large sense, the types of hate speech covered by extant datasets can be classified into two categories: target-specific and target-generic hate speech. First, target-specific hate speech has a deliberate target for hatred, including religion (Zannettou et al., 2020), race (Waseem and Hovy, 2016; Basile et al., 2019) and gender (Fersini et al., 2020). On the contrary, target-generic hate speech encompasses a broad range of hatred without a specific target group, such as bullying or abusing random people (Chatzakou et al., 2017).

The vast majority of hate speech datasets are developed in English (Poletto et al., 2021). However, there is a growing interest in increasing

---

[1] In this context, a comment thread refers to a discussion consisting of a main or top-level comment along with multiple replies or responses to that comment.
[2] https://github.com/jnnpk/hate_instiga ting_speech_dataset

[3] We use the term hate instigator at a text level, not at a human level. That is, it refers to a hate instigating comment, not a human who incite hatred of others.

the linguistic diversity of hate speech datasets (Chiril et al., 2022). For instance, Van Hee et al. (2015) have suggested a Dutch dataset for detecting cyberbullying on a social media site. Ross et al. (2017) have collected German tweets representing hatred towards refugees. Sanguinetti et al. (2018) have developed a dataset comprising Italian tweets labeled with the different levels of offensiveness and aggressiveness. Moon et al. (2020) have created a Korean dataset consisting of comments exhibiting social bias and hatred.

The main data source of existing hate-speech datasets has been Twitter (Wassem and Hovy, 2016; Davidson et al., 2017). On top of that, other social media sites, including Facebook (Rodriguez et al., 2019), Yahoo! (Nobata et al., 2016), Reddit (Qian et al., 2019), Instagram (Corazza et al., 2019) have been used as important sources to collect data from. In addition, video sharing websites, such as YouTube, have been recognized as valuable data sources that prior researchers have extensively explored (Sharma et al., 2018; Pavlopoulos et al., 2017). While these studies have contributed to a comprehensive collection of datasets for the development of automated hate speech detectors, as stated earlier, there has been a limited emphasis on creating datasets specifically tailored for identifying hate instigating speech.

## 2.2 Hate instigating speech

Hate speech has a higher spreading velocity and it tends to spread farther and wider compared to normal posts (Mathew et al., 2019). Hate speech infects other people to spread even more hatred in the society, thriving like a disease (Ahammed et al., 2019). However, prior studies, particularly those that use ML as a moderation tool, have only considered hate speech itself rather than its possibility of influencing others (Mondal et al., 2018). A few studies, not necessarily in the field of ML-based hate speech detection, have made exceptions to this trend. For instance, ElSherief et al. (2018) suggested a concept of *hate instigators* to delve into the underlying causes of hate behaviors. They defined hate instigators as individuals who actively engage in expressing explicit and targeted hatred, thereby triggering the occurrence of hate behaviors by others. However, their definition may have limitations since it assumes that individuals who are classified as hate instigators always present hatred behaviors, which might not always be the case. Furthermore, the

detection of hate instigation at the individual level introduces additional concerns including issues related to privacy and ethics. For example, in the development of a hate instigation detection model aimed at individuals, it becomes necessary to gather user-specific data, potentially raising privacy concerns. Also, directing moderation efforts towards individuals (e.g., blocking user accounts), rather than hate instigation speech (e.g., removing comments), may give rise to ethical concerns. Therefore, in this study, we consider hate instigation at a behavior level (i.e., text level) rather than at an individual level.

Similar to our focus, there have been several recent studies examining hate instigation detection at a behavioral level. For example, Sahnan et al. (2021) and Dahiya et al. (2021) have provided datasets that can be used to predict the hate intensity of replies given a tweet. However, for the development of the datasets, both studies have computational approaches for estimating hate intensity scores whose results can diverge from the actual ground truth. On the other hand, our dataset is meticulously curated through manual labeling, which yields more accurate results of labeling. Furthermore, our dataset is more diverse in that it is multilingual (i.e., English and Korean) and its source encompasses YouTube channels biased towards both ends of the political spectrum (i.e., Democrats and Republicans). In the next section, we describe the details of our dataset and its collection process.

## 3 Method

### 3.1 Data collection and annotation

YouTube has emerged as a valuable source for numerous datasets developed in the field of hate speech detection. This is primarily due to the presence of political discussions and conflicts that frequently occur on the platform. Hence, in order to construct a comprehensive dataset of hate instigating speech, we collected comments data from YouTube channels operated by major news media companies in the US and South Korea. Specifically, our list includes CNN, Fox News, JTBC, and TV Chosun, which allows us to capture a diverse range of political perspectives in both English (left-leaning: CNN; right-leaning: Fox News) and Korean (left-leaning: JTBC; right-leaning: TV Chosun). The development of our dataset primarily centers around the identification

of the impact of a comment on its corresponding replies. That is, we aim to investigate how comment-level texts, as illustrated in Figure 1-B, impact the responses and reactions of individuals who engage in replying to the original comment. Depending on the level of hatred presented in these replies, we may be able to understand whether the comment instigates hatred responses or not (i.e., whether the comment is hate instigating speech or not).

We collected data using YouTube Data API[4], the comments and replies of the videos posted on the aforementioned list of YouTube channels during a timeframe between January 2022 and April 2022. As a result, we could collect in total 4165 comments and 11310 replies on 210 videos. Specifically, CNN had 46 videos, 935 comments, and 2499 replies; FoxNews had 44 videos, 980 comments, and 2592 replies; JTBC had 45 videos, 1018 comments, and 3496 replies; and TVChosun had 75 videos, 1232 comments and 2723 replies.

As previously mentioned, our main objective is to create a dataset that includes annotations indicating whether comments are hate instigating speech or not. The annotation process for our dataset involves two key steps. First, we label the replies associated with each comment as either hate speech or non-hate speech. This step helps us understand the nature of the responses generated by users in relation to the original comment. Secondly, we measure the level of hate speech instigation at the comment level by examining the ratio of hate replies associated with each comment (i.e., the proportion of hateful replies received in relation to the total number of replies). This metric allows us to gauge the extent to which a comment may incite or provoke hate speech from others.

For a consistent annotation, we draw upon existing literature (Assimakopoulos et al., 2020; Mulki et al., 2019; Pavlopoulos et al., 2017; Davidson et al., 2017; Mathur et al., 2018; Sigurbergsson and Derczynski, 2019) to define hate speech and non-hate speech as follows:

(a) Hate Speech: The speech that deliberately targets a specific group or individual for condemnation. This includes, but is not limited to, sexism, homophobia, and hate towards certain political groups. Additionally, speech that is aggressive but does not have a specific target also falls under this category.

(b) Non-Hate Speech: The speech that does not fall under the above categories. Criticism and sarcasm do not qualify as hate speech.

With these definitions in place, three graduate-level students with expertise in the domain of hate speech detection participated in the annotation process. To ensure consistency among annotators, we have developed detailed guidelines that provide a clear and comprehensive definition of hate speech. These guidelines outline specific criteria and examples to help annotators accurately identify instances of hate speech. They cover various aspects, such as explicit and implicit forms of hate speech, targeted groups or individuals, offensive language, discriminatory content, and other relevant factors. The guidelines aim to provide a common understanding and interpretation of hate speech, ensuring that all annotators follow a consistent approach when labeling comments and replies in the dataset.

In order to refine and validate our annotation guidelines, we conducted a pilot-annotation process. During this phase, a subset of the dataset (i.e., 100 samples) was randomly selected, and a team of annotators applied the provided guidelines to label given data. The pilot annotation allowed us to assess the clarity and effectiveness of the guidelines and identify any potential issues or ambiguities. After completing the pilot annotation, we conducted a thorough review and analysis of the annotated data to ensure consistency and agreement among the annotators. Any discrepancies or areas of uncertainty were discussed and addressed through further refinement of the guidelines.

Based on the refined annotation guidelines, our team of annotators proceeded to label the entire dataset. To statistically measure the level of agreements among the annotators, we calculated Fliess' kappa for each subset of data: CNN = 0.586; FoxNews = 0.425; JTBC = 0.340; and TVChosun = 0.610. We argue that these values indicate our annotation process has demonstrated a good level of agreement (Gisev et al., 2013).

[4]https://developers.google.com/youtube/v3

## 3.2 Data description

To gain initial insights into the characteristics of hate instigating speech, we conduct an explorative analysis, offering an overview of our annotated data and shedding light on various patterns of hate instigating speech. First, in Table 1, we show descriptive statistics at a comment level for each language in our dataset. Note that, as mentioned earlier, we gauge the level of hate instigated by a comment in a continuous manner (i.e., the ratio of hate replies to total replies). However, in this section, we apply a binary classification approach where a comment is classified as HIS if it has at least one reply associated with it.

In the case of English, our dataset consists of a total of 976 instances of hate instigating speech (henceforth, referred to as HIS) and 939 instances of non-hate-instigating speech (henceforth, referred to as NHIS). The average number of words in each comment was 40.26 for HIS and 41.07 for NHIS. On top of that, we also calculated the average number of unique words in a comment (i.e., vocabulary size of a comment). We noticed that a slightly larger number of unique words were used in NHIS comments compared to HIS comments (i.e., 27.79 and 27.04, respectively). In addition, we observed that HIS received a higher level of attention. That is, on average, 3.03 replies were associated with one HIS comment while only 2.77 replies were associated with one NHIS comment. Among the replies to HIS comments, approximately 63.0% contained hatred (i.e., hate replies).

On the other hand, our Korean dataset consists of a total of 861 instances of HIS and 1,389 instances of NHIS. The average number of words in each comment was 18.57 for HIS and 22.02 for NHIS. In addition, for HIS, on average, 14.18 unique words were used while, for NHIS, a much larger number of unique words (i.e., 17.14) were used, confirming the pattern of vocabulary usage that we identified from the English dataset. On top of that, another similar trend could be found regarding the amount of attention given to HIS and NHIS. That is, HIS comments received a much larger attention than NHIS comments, with an average of 3.89 replies compared to 2.08 replies. Among the replies to HIS comments, approximately 48.3% contained hatred (i.e., hate replies)

In Table 2, we further break down the information in Table 1 by media channel (i.e., news

| | English | | Korean | |
|---|---|---|---|---|
| | **HIS** | **NHIS** | **HIS** | **NHIS** |
| Total Number of Comments | 976 | 939 | 861 | 1389 |
| Average Number of Words in Each Comment* | 40.26 (88.75) | 41.07 (81.99) | 18.57 (18.63) | 22.02 (34.64) |
| Average Vocabulary Size of Each Comment* | 27.04 (36.49) | 27.79 (33.26) | 14.18 (13.33) | 17.14 (22.26) |
| Average Number of Replies* | 3.03 (4.27) | 2.77 (8.65) | 3.89 (5.47) | 2.08 (3.03) |
| Average Number of Hate Replies* | 1.91 (2.26) | N/A | 1.88 (1.77) | N/A |
| * Standard deviations are noted in parentheses. | | | | |

Table 1: Descriptive Statistics of HIS and NHIS by Language.

publisher). One of the interesting findings is that, in both US and Korea, the politically right-leaning media channels (i.e., FoxNews and TVChosun) had a proportion of HIS to NHIS. In addition, the proportion of hate replies to total replies associated with HIS was also higher in the right-leaning channels.

One interesting aspect of our data is that, unlike prior studies, we focus on the nature of speech that can provoke hate, regardless of whether the speech itself includes hatred. That is, while prior studies in general have only emphasized the propagation of hatred through a chain of hate speech, we claim that even non-hate speech can be the cause of hatred (Anderson and Barnes, 2022; Khurana et al., 2022). For instance, in Table 3, we further categorize HIS into hate HIS and non-hate HIS. The former indicates HIS comments that contain expressions of hatred while the latter refers to HIS comments but do not necessarily contain expressions of hatred. According to Table 3, we identify that a large proportion of HIS comments are in fact non-hate HIS. In addition, they induce a higher level of attention in both languages (i.e., 3.07 replies compared to 2.82 replies in English; 4.01 replies compared to 2.76 replies in Korean). This notable

|  | CNN | | Fox News | | JTBC | | TV Chosun | |
|---|---|---|---|---|---|---|---|---|
|  | **HIS** | **NHIS** | **HIS** | **NHIS** | **HIS** | **NHIS** | **HIS** | **NHIS** |
| Total Number of Comments | 418 | 517 | 558 | 422 | 209 | 809 | 652 | 580 |
| Average Number of Words in Each Comment* | 53.10 (128.52) | 49.30 (105.86) | 30.65 (34.49) | 30.99 (32.30) | 18.94 (14.96) | 22.95 (28.12) | 18.45 (19.67) | 20.74 (42.06) |
| Average Vocabulary Size of Each Comment* | 32.88 (49.88) | 32.13 (40.74) | 22.66 (20.58) | 22.47 (19.47) | 15.17 (11.47) | 18.24 (19.41) | 13.86 (13.87) | 15.60 (25.67) |
| Average Number of Replies* | 2.99 (4.51) | 3.12 (10.67) | 3.07 (4.08) | 2.25 (4.52) | 6.97 (8.42) | 2.52 (3.79) | 2.88 (3.54) | 1.48 (1.15) |
| Average Number of Hate Replies* | 1.74 (2.12) | *N/A* | 2.03 (2.35) | *N/A* | 1.52 (1.23) | *N/A* | 2.03 (1.85) | *N/A* |

\* Standard deviations are noted in parentheses.

Table 2: Descriptive Statistics of HIS and NHIS by Media Channel.

|  | English | | Korean | |
|---|---|---|---|---|
|  | **Hate HIS** | **Non-hate HIS** | **Hate HIS** | **Non-hate HIS** |
| Total Number of Comments | 255 | 721 | 111 | 750 |
| Average Number of Words in Each Comment* | 39.10 (64.61) | 40.67 (95.85) | 19.46 (18.40) | 18.43 (18.67) |
| Average Vocabulary Size of Each Comment* | 26.56 (29.82) | 27.21 (38.59) | 15.24 (13.39) | 14.02 (13.32) |
| Average Number of Replies* | 2.82 (4.06) | 3.07 (4.32) | 2.76 (3.64) | 4.01 (5.65) |
| Average Number of Hate Replies* | 2.05 (2.69) | 1.85 (2.08) | 1.36 (0.81) | 1.83 (1.98) |

\* Standard deviations are noted in parentheses.

Table 3: Descriptive Statistics of Hate HIS and Non-hate HIS by Language.

result highlights the limitation of existing hate speech moderation approaches, which mostly focus on hate speech itself but not its root. It is highly likely that they may fail to effectively manage non-hate HIS which could significantly contribute to the propagation of hatred. Therefore, we claim that it is particularly important to shift the paradigm of hate speech moderation to hate instigating speech moderation, and our study is one of the pioneering efforts in this direction. In the next section, we show the practical application of our dataset in combination with extant ML methods for effectively managing hate instigating speech.

## 4 Data application

### 4.1 Experimental setting

In this section, we present how our dataset can be applied in a real-world setting. We develop and evaluate a computational model for detecting hate instigation speech using multiple off-the-shelf ML methods. Specifically, our problem setting aims to develop ML models that accurately predict the level of hate instigation based on text data. Formally, it is to identify an optimal function, $f: (x_i) = [0,1]$ , where $x_i$ is an element of the potential HIS set $X = \{x_1, x_2, \cdots, x_n\}$. The range of function $f$ is a closed interval between 0 and 1, which signifies the likelihood of a given text being classified as HIS (a higher value indicates a higher probability of the text being classified as HIS).

As mentioned earlier, we tested ML methods that are largely adopted in practice, including linear regression (i.e., Lasso regression), tree-based models (i.e., random forest, extra tree, and gradient boosting; Davidson et al., 2017), and multiple variations of neural networks (i.e., CNN, LSTM, and BERT Regressor; Wolf et al., 2019), to develop computational models for the HIS detection. For preprocessing of texts and their projection to a vector space, we used Spacy[5], an open-source software library for natural language processing. For each baseline, we trained and tested it based on five-fold cross validation. It is important to note that our primary objective in this study is to showcase the applicability of our dataset. As a result, we did not engage in an extensive process of hyperparameter tuning. The focus was on demonstrating the feasibility and potential of our dataset, rather than achieving optimal performance through fine-tuning of hyperparameters.

To assess the performance of the baselines, we adopted the Mean Absolute Error (MAE) as the primary evaluation metric, calculating the absolute difference between the predicted probability scores and true scores. This aligns with our objective of hate instigator detection, where accurately estimating the probability of texts instigating hatred is significant.).

## 4.2  Results

Table 4 serves as a summary of the results of the aforementioned experiment, offering insights into the effectiveness of different ML methods in identifying HIS. There are several interesting points to be made from it. First, it appears that the difficulty level of detecting HIS varies depending on the language being analyzed. That is, in general, the task of detecting HIS in English produces a higher level of error rate than in Korean. This result remains the same even when a pretrained language model, which predominantly leverages English corpora over Korean corpora[6], is used as a baseline (i.e., BERT Regressor). To find an explanation for the discrepancy in difficulty levels between the two languages, we conducted a manual exploration of subsets of English and Korean HIS comments. In English, HIS comments encompassed subtle cases, such as sarcastic and satirical expressions, which may require nuanced understanding to identify as instances of hate speech. In contrast, Korean HIS comments tended to be more direct, often containing offensive or explicit language that was readily recognizable as offensive. The variation in language-specific difficulties underscores the importance of considering language-specific factors when developing detection methods or models for HIS.

On top of that, except for the JTBC dataset, off-the-shelf ML methods were not effective in identifying HIS comments. That is, the results regarding CNN, FoxNews, and TVChosun yielded a high level of MAE, deviating from our initial expectation. The difficulty of predicting hate instigation speech based on extant ML methods aligns with the results of the previous studies within a similar context (Dahiya et al., 2021). This

| | (1) CNN | (2) FoxNews | English (1+2) | (3) JTBC | (4) TVChosun | Korean (3+4) |
|---|---|---|---|---|---|---|
| Lasso Regression | 0.416 | 0.426 | 0.428 | 0.179 | 0.415 | 0.347 |
| Gradient Boosting | 0.413 | 0.430 | 0.421 | 0.188 | 0.417 | 0.342 |
| Extra Trees | 0.414 | 0.419 | 0.425 | 0.193 | 0.417 | 0.343 |
| Random Forest | 0.416 | 0.424 | 0.421 | 0.192 | 0.420 | 0.348 |
| SVM | 0.401 | 0.431 | 0.430 | 0.210 | 0.412 | 0.315 |
| CNN | 0.364 | 0.491 | 0.467 | 0.131 | 0.428 | 0.252 |
| LSTM | 0.364 | 0.458 | 0.469 | 0.131 | 0.400 | 0.258 |
| BERT Regressor | 0.362 | 0.434 | 0.469 | 0.128 | 0.410 | 0.245 |

Table 4: Performance Comparison of HIS Detection Models by Media Channel.

---

[5] https://spacy.io/

[6] With respect to the development of pre-trained language models for difference purposes, English corpora have been the main source of data while other languages, including Korean, have been paid less attention (Kim et al., 2022).

This discrepancy in attention and resources allocated to different languages has led to a relative imbalance in the availability and quality of pre-trained language models for non-English languages.

indicates that there are a higher level of challenges and complexities associated with identifying HIS than initially expected. This calls for future research to delve into the specific linguistic nuances of HIS and to develop ML methods that can effectively capture and address these nuances.

## 5 Discussion

In this study, we provide a novel concept of hate instigation speech (HIS) and develop a dataset that can be a catalyst for future research on hate speech moderation, particularly those based on ML-based approaches. Compared to prior studies, our approach of highlighting HIS, which identifies and manages the root of hate speech, is a more effective way of dealing with the propagation of online hate speech. That is, by targeting the root causes of hatred, we can potentially have a greater impact on reducing the propagation and harmful effects of hate speech in online environments. Furthermore, our approach is capable of addressing broader forms of discourse, which may contribute to the propagation of hate speech but were not the primary focus of moderation in previous studies. That is, while non-hate HIS does not explicitly manifest hatred by itself but still significantly contributes to the provocation of hostility, aggression, and toxicity, prior studies have not paid proper attention to it. By introducing such a broader concept of harmful discourse, we expect more comprehensive moderation strategies which will foster healthier and more inclusive online environments.

## 6 Limitations

While we expect that our study will make significant contributions to the field of hate speech moderation, it is not without limitations. First, one potential limitation is that we did not consider the chain of hate instigation. That is, we assumed that all replies are affected by the comment associated with it. However, there might be some cases where a reply, rather than a comment, affects other replies. Nonetheless, as the pioneering effort to create a human-annotated dataset for identifying hate-instigating speech provides a foundational basis for future studies that can potentially present a more nuanced and precise definition of this phenomenon. In addition, there is a possibility that our definition of HIS is too broad or coarse, warranting a more nuanced and granular approach. Instead of simply classifying texts as either HIS or NHIS, we may consider the multifaceted nature of hate instigating speech. This will involve categorizing HIS into various subcategories that correspond to different types of hatred, such as racism instigating speech, sexism instigating speech, religious HIS, motiveless HIS, and so on. Lastly, it is worth nothing that our data in this study solely comprises textual information. However, incorporating a different modality of data, such as images, may provide a different angle to our problem setting.

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
