# OpenReview forum: "Uncovering the Root of Hate Speech: A Dataset for Identifying Hate Instigating Speech"
_EMNLP/2023/Conference — EMNLP 2023 Findings_

### Official Review · Reviewer_wEu5 · 2023-07-31

**Typos Grammar Style And Presentation Improvements:** N/A
**Soundness:** 1

**Excitement:**

1: Poor: I cannot identify the contributions of this paper, or I believe the claims are not sufficiently backed up by evidence. I would fight to have it rejected.

**Missing References:**

N/A

**Paper Topic And Main Contributions:**

The authors introduce the concept of Hate Instigating Speech (HIS), a specific type of textual posts on online platforms (Youtube) that stimulate or provoke others to engage in hate speech. The paper highlights the importance of targeting the root cause of hate speech rather than focusing solely on individual instances of hate speech. The authors provide a multilingual (English and Korean) dataset designed for HIS.

**Questions For The Authors:**

N/A

**Reasons To Accept:**

While urging interest in hate instigating speech, the authors have constructed a multilingual dataset for this purpose.

**Reasons To Reject:**

1. The criteria for Hate Instigating Speech are not clear. Merely categorizing a post as Hate Instigating Speech in the comments on that post lacks persuasiveness.
2. As the key contribution of this paper lies in dataset construction, it is essential to provide a more detailed explanation of the criteria and guidelines for annotation.


**Reproducibility:**

3: Could reproduce the results with some difficulty. The settings of parameters are underspecified or subjectively determined; the training/evaluation data are not widely available.

**Reviewer Confidence:**

4: Quite sure. I tried to check the important points carefully. It's unlikely, though conceivable, that I missed something that should affect my ratings.

---

> ### Author Rebuttal · Authors · 2023-08-28
>
> Thank you for your feedback. Below, we provided point-by-point responses to your comments.
>
> Our response to your comment #1: We concur with your feedback that our definition of hate-instigating speech has some limitations such as not considering the dynamics between replies. For instance, a reply, instead of a comment, can be the cause of hate presented in another reply. Nonetheless, given that our work represents the initial attempt to establish a human-annotated dataset for identifying hate-instigating speech, we contend that our study serves as a stepping stone for future research, which may potentially present a more nuanced and precise definition of hate-instigating speech.
>
> Our response to your comment #2: In the manuscript (Section 3.1.), we did include our annotation process in general but we found out that there was a lack of definitions regarding hate and non-hate speech. We provide them below.
>
> <Definitions of hate speech and non-hate speech>
> Hate Speech: Speech that deliberately targets a specific group or individual for condemnation. This includes, but is not limited to, sexism, homophobia, and hate towards certain political groups. Additionally, speech that is aggressive but does not have a specific target also falls under this category.
>
> Examples:
> Explicit hatred towards a particular group: "INDIA + CHINA = DISGUSTING RAT EATING 3RD WORLD COUNTRIES!"
> Hate towards a political group: "Biden Is the biggest Disaster in American History 😫"
> Sexism: "She is a woman; it's hard for women to understand things like this."
> General aggression: "YouTube comments are such a cesspit."
> Non-Hate Speech: Speech that does not fall under the above categories. Criticism and sarcasm do not qualify as hate speech in our dataset.
>
> Examples:
> Sarcasm: "Wow, what a surprise, American."
> Criticism: "I'm part Russian and absolutely disappointed in this whole war!"
>
> The definitions and examples are based on prior literature (Assimakopoulos et al., 2020; Davidson et al., 2017; Mulki et al., 2019; Pavlopoulos et al., 2017).
>
> Assimakopoulos, S., Muskat, R. V., Van Der Plas, L., & Gatt, A. (2020). Annotating for hate speech: The MaNeCo corpus and some input from critical discourse analysis. arXiv preprint arXiv:2008.06222.
>
> Davidson, T., Warmsley, D., Macy, M., & Weber, I. (2017, May). Automated hate speech detection and the problem of offensive language. In Proceedings of the international AAAI conference on web and social media (Vol. 11, No. 1, pp. 512-515).
>
> Mulki, H., Haddad, H., Ali, C. B., & Alshabani, H. (2019, August). L-hsab: A levantine twitter dataset for hate speech and abusive language. In Proceedings of the third workshop on abusive language online (pp. 111-118).
>
> Pavlopoulos, J., Malakasiotis, P., & Androutsopoulos, I. (2017, September). Deeper attention to abusive user content moderation. In Proceedings of the 2017 conference on empirical methods in natural language processing (pp. 1125-1135).

---

### Official Review · Reviewer_eite · 2023-08-05

**Soundness:** 2

**Excitement:**

2: Mediocre: This paper makes marginal contributions (vs non-contemporaneous work), so I would rather not see it in the conference.

**Paper Topic And Main Contributions:**

This paper looks at the problem of hate speech instigation to determine the underlying cause of hate speech. The authors collect data from multiple news websites in English and Korean to establish this problem.

**Questions For The Authors:**

1. Were samples to create this dataset scraped from news websites?

**Reasons To Accept:**

1. The paper looks at an overlooked angle of dialogs for hate speech, hence dealing with hate speech instigation.

**Reasons To Reject:**

1.	It is unclear by which definition of hate speech (fig1) the replies are hate speech. Offensive words may be toxic, but may not constitute hate speech.
2.	(L217) Behaviour level is also not always the case. One can argue anything influential can be a hate instigator.
3. Models used are not the best-in-class, yet the conclusion of existing models being insufficient to detect hate speech instigation is claimed.

**Reproducibility:**

3: Could reproduce the results with some difficulty. The settings of parameters are underspecified or subjectively determined; the training/evaluation data are not widely available.

**Reviewer Confidence:**

4: Quite sure. I tried to check the important points carefully. It's unlikely, though conceivable, that I missed something that should affect my ratings.

---

> ### Author Rebuttal · Authors · 2023-08-28
>
> Thank you for your feedback. Below, we provide our point-by-point responses regarding your comments.
>
> Our response to your comment #1. Sorry for the confusion. In Figure 1, our intention was to underscore the core emphasis of our study. While previous investigations (1-(A)) focus on identifying hate speech without considering the hierarchical structure of comments and replies, our research (1-[B]) centers on comments, particularly their potential to incite or provoke hate speech at the reply level. This distinction holds significance because identifying hate-instigating speech can prove more effective in curbing the proliferation of hate speech compared to the blanket detection of all instances.
>
> In addition, our definition of hate speech is based on prior studies. For instance, Davidson et al. (2017) defined hate speech in a broad sense as "language that is used to express hatred towards a targeted group or is intended to be derogatory, to humiliate, or to insult the members of the group." As you have mentioned, at the word level, we cannot determine whether a comment contains hate or not. Therefore, we centered our definition of hate speech at the comment (or reply) level and employed the human annotation process instead of the machine annotation process since the latter, often based on lexicon-based models, could fail in accurately identifying hate speech.
>
> Our response to your comment #2. We concur that various factors, including individual characteristics, can indeed impact the likelihood of producing hate-instigating speech. While it would be ideal to possess such information, the practical reality is that identifying personal characteristics from a mere snippet of information is nearly impossible. This is precisely why our focus rests on behavioral aspects, particularly linguistic characteristics associated with hate speech.
>
> Our response to your comment #3. We agree with your observation that the models employed in this study are not best-in-class. However, the primary focus of this study is on the novelty of the dataset provided rather than the methods employed. In addition, there is empirical evidence from prior literature showing that even state-of-the-art language models often fail to solve similar tasks, highlighting the difficulties of identifying hate-instigating speech (Dahiya et al., 2021). We assert that our dataset has the potential to serve as a stepping stone for future studies aimed at overcoming these difficulties.
>
> Dahiya, S., Sharma, S., Sahnan, D., Goel, V., Chouzenoux, E., Elvira, V., ... & Chakraborty, T. (2021, August). Would your tweet invoke hate on the fly? forecasting hate intensity of reply threads on twitter. In Proceedings of the 27th ACM SIGKDD Conference on Knowledge Discovery & Data Mining (pp. 2732-2742).
> Davidson, T., Warmsley, D., Macy, M., & Weber, I. (2017, May). Automated hate speech detection and the problem of offensive language. In Proceedings of the international AAAI conference on web and social media (Vol. 11, No. 1, pp. 512-515).

---

### Official Review · Reviewer_9nDB · 2023-08-06

**Typos Grammar Style And Presentation Improvements:** 1. Line 387 typo
**Soundness:** 2

**Excitement:**

2: Mediocre: This paper makes marginal contributions (vs non-contemporaneous work), so I would rather not see it in the conference.

**Missing References:**

Enlisted the papers above.

**Paper Topic And Main Contributions:**

The paper aims to establish the importance of studying root posts that may not be classified as hate speech via text classification systems but are nevertheless hate instigating in nature. The authors curate such a dataset from Youtube in English and Korean and then compare the modelling of the said dataset via ML-based regressors.

**Reasons To Accept:**

1. Introduce a new dataset in the hate speech literature.
2. Bring back the focus on dataset that is not only about hate speech detection but rather its propagation (aka instigation)

**Reasons To Reject:**

1. While the attempts to present a new dataset on hate instigation speech, the information presented is not complete and accurate because:
Existing literature needs to be appropriately cited and compared. There are existing datasets in English that attempt to determine the instigation level of a post, work: Better Prevent than React: Deep Stratified Learning to Predict Hate Intensity of Twitter Reply Chains", "Would your tweet invoke hate on the fly? Forecasting hate intensity of reply threads on Twitter", "Ruddit: Norms of Offensiveness for English Reddit Comments.", "Proactively Reducing the Hate Intensity of Online Posts via Hate Speech Normalization" needs to be examined in this study.
2. Being a dataset paper, it needs to provide the following details.
2a: Did the authors only look at political content on the news channels?
2b: Was any text preprocessing applied before and after the annotation process?
2c: Information on the annotation guidelines, definitions used and finalized, and the annotation process must be included in a detailed and systematic manner.
2d: Demographic details about the annotators and how the disagreements were handled should also be included.
2e:The authors should also highlight if, at first, the annotations were for only hate and non-hate and then mapped to HIS and NHIS. How many posts from each category map to the other should also be enlisted in a table.
3. In the dataset description section, authors ought to substantiate their analysis with significance testing when they state that HIS speech received more comments and more hateful comments (line 386).
4. Calling this dataset a hate instigation dataset may not be correct, as the dataset is not looking at instigation from an underlying social-physio perspective but instead just looking at posts that are more likely to receive hateful posts (hate prone/hate provoking rather than hate instigation would be better terminology.)

**Reproducibility:**

3: Could reproduce the results with some difficulty. The settings of parameters are underspecified or subjectively determined; the training/evaluation data are not widely available.

**Reviewer Confidence:**

4: Quite sure. I tried to check the important points carefully. It's unlikely, though conceivable, that I missed something that should affect my ratings.

---

> ### Author Rebuttal · Authors · 2023-08-28
>
> Thank you for your insightful comments. We appreciate the opportunity to clarify and strengthen our work in light of your feedback. Below, we provide our point-by-point responses regarding your comments.
>
> Our response to your comment #1: We acknowledge the importance of situating our work within the context of existing literature. The papers you mentioned, are indeed relevant to the broader context of our study. However, while there are certain similarities between the studies you mentioned and our work, it's important to note that there exist notable distinctions as well.
>
> First, the studies you mentioned (Sahnan et al. [2021] and Dahiya et al. [2021]), both using the same dataset, primarily relied on machine-learning-based and lexicon-based computational techniques for estimating hate intensity scores, and hence their results can diverge from the actual ground truth. In contrast, our dataset is meticulously curated through manual labeling, which tends to yield more accurate results of labeling.
>
> Furthermore, while Sahnan et al. (2021) and Dahiya et al. (2021) focused solely on political events, our dataset encompasses a diverse range of news domains, including politics, economics, and various social subjects. This broader scope provides a comprehensive perspective that distinguishes our work.
>
> Moreover, our dataset stands out as bilingual, incorporating both English and Korean, whereas the other studies are confined to English only. This linguistic inclusivity contributes to the richness and applicability of our dataset.
>
> The unique collection platform is another aspect that differentiates our study. Our data is sourced from YouTube, which introduces distinctive environmental factors that can influence hate-instigating behaviors, setting it apart from the platforms that other studies have collected data.
>
> Lastly, it's important to highlight that the research objectives of Hada et al. (2021) and Masud et al. (2022) – the two other studies you mentioned – diverge from our focus on hate instigation. Specifically, Masud et al. (2022) introduce a method for capturing specific hate-related parts within a text, while Hada et al. (2021) propose a novel approach for annotating levels of hate speech. These divergent objectives underscore the unique contribution of our study, centered on hate instigation.
>
> In our revised manuscript, we will ensure to incorporate these references to provide a more comprehensive overview of the existing literature, further underscoring the distinct value and complementarity of our research in the domain of hate-instigating speech prediction.
>
> Our responses to your comment #2:
> 2a: Our dataset includes content from diverse news channels such as CNN, FOX News, JTBC, and Chosun. While it is true that these platforms often focus on political discussions, our dataset is not limited to political content as mentioned in our response to your comment #1. This ensures that our dataset captures a wide range of hate-instigating speech across diverse domains.
> 2b: We did not apply any text preprocessing to the crawled data before the annotation process. The data was used in its raw form to maintain the original context and nuances, which we believe is crucial for the quality of the annotations.
> 2c: We acknowledge the importance of providing a detailed and systematic explanation of our annotation guidelines, definitions, and processes. In the manuscript, we did include our annotation process in general but we found out that there was a lack of definitions regarding hate and non-hate speech. We provide them below.
>
> <Definitions of hate speech and non-hate speech>
>
> Hate Speech: Speech that deliberately targets a specific group or individual for condemnation. This includes, but is not limited to, sexism, homophobia, and hate towards certain political groups. Additionally, speech that is aggressive but does not have a specific target also falls under this category.
>
> - Examples:
> - Explicit hatred towards a particular group: "INDIA + CHINA = DISGUSTING RAT EATING 3RD WORLD COUNTRIES!"
> - Hate towards a political group: "Biden Is the biggest Disaster in American History 😫"
> - Sexism: "She is a woman; it's hard for women to understand things like this."
> - General aggression: "YouTube comments are such a cesspit."
>
> Non-Hate Speech: Speech that does not fall under the above categories. Criticism and sarcasm do not qualify as hate speech in our dataset.
> - Examples:
> - Sarcasm: "Wow, what a surprise, American."
> - Criticism: "I'm part Russian and absolutely disappointed in this whole war!"
>
> The definitions and examples are based on prior literature (Assimakopoulos et al., 2020; Mulki et al., 2019; Pavlopoulos et al., 2017; Davidson et al., 2017; Mathur et al., 2018; Sigurbergsson and Derczynski, 2019).
>
> 2d: The annotations were performed by three graduate-level students. Disagreements among annotators were resolved through discussions and by achieving a high Fleiss-alpha score, as indicated in line 336 of our manuscript.
>
> 2e: As described in line 287 and line 353 of our manuscript, our annotation process involves two key steps. Initially, replies are labeled as either hate speech or non-hate speech. Subsequently, we measure the level of hate instigation at the comment level based on the proportion of hateful replies. Comments are classified as hate-instigating speech if they have at least one associated hateful reply.
>
> Our response to your comment #3: To clarify, we observed that hate-instigating speech (HIS) in our dataset received more comments overall, while non-hate-instigating speech (NHIS) received fewer comments. Additionally, by definition, comments categorized as NHIS received no hateful replies. Our intention in stating these observations was not to argue statistical differences between the two groups of comments but to provide a descriptive analysis of the observed trends in our dataset. We will make this point explicit in the revised manuscript to avoid any potential confusion.
>
> Our response to your comment #4: While we acknowledge that the dataset does not delve into the underlying social-physiological aspects of hate instigation, the focus of this paper is rather methodological, aimed at presenting the dataset itself. The term "hate-instigating speech" was chosen to emphasize the dataset's unique focus on speech that is more likely to elicit hateful responses. However, we appreciate your suggestion to consider alternative terminology like "hate prone" or "hate provoking." We will take this into account in the revision process.

---

### Meta-Review · Area_Chair_KiEx · 2023-09-19

**Recommendation:** 3

**Metareview:**

This paper introduces new dataset for hate speech instigation. Reviewers raised concern about the missing literature, annotations, and modeling. Authors in their rebuttal have provided reasonable explanation for these points and how these could be addressed.

---

### Decision · Program_Chairs · 2023-10-07

**Decision:**

Accept-Findings

**Comment:**

This paper introduces new dataset for hate speech instigation. Reviewers raised concern about the missing literature, annotations, and modeling. Authors in their rebuttal have provided reasonable explanation for these points and how these could be addressed.